# Phytochemical Composition, Antioxidant, and Antimicrobial Attributes of Different Solvent Extracts from *Myrica esculenta* Buch.-Ham. ex. D. Don Leaves

**DOI:** 10.3390/biom9080357

**Published:** 2019-08-09

**Authors:** Atul Kabra, Rohit Sharma, Christophe Hano, Ruchika Kabra, Natália Martins, Uttam Singh Baghel

**Affiliations:** 1Research Scholar, I.K. Gujral Punjab Technical University, Kapurthala 144603, Punjab, India; 2Department of Pharmacology, Kota College of Pharmacy, Kota 325003, Rajasthan, India; 3Central Ayurveda Research Institute for Drug Development, CCRAS, Ministry of AYUSH, Government of India, Bidhannagar, Kolkata 700091, West Bengal, India; 4Laboratoire de Biologie des Ligneux et des Grandes Cultures, INRAUSC1328, Universitéd’Orléans, 45100 Orléans, France; 5Department of Pharmaceutical Chemistry and Analysis, Kota College of Pharmacy, Kota 325003, Rajasthan, India; 6Faculty of Medicine, University of Porto, 4200-319 Porto, Portugal; 7Institute for Research and Innovation in Health (i3S), University of Porto, 4200-135 Porto, Portugal; 8Department of Pharmacy, University of Kota, Kota 325003, Rajasthan, India

**Keywords:** *Myrica esculenta*, antioxidant, antimicrobial, phenolic composition

## Abstract

*Background*: Plant diversity is a basic source of food and medicine for local Himalayan communities. The current study was designed to assess the effect of different solvents (methanol, ethyl acetate, and water) on the phenolic profile, and the corresponding biological activity was studied. *Methods*: Antioxidant activity was investigated using 2,2-diphenyl-1-picrylhydrazyl (DPPH) and 2,2″-azino-bis(3-ethylbenzothiazoline-6-sulphonic) acid (ABTS) assay, while the antimicrobial activity was evaluated by disk diffusion method using various bacterial and fungal strains. *Results*: The outcomes demonstrated that methanol acted as the most effective solvent for polyphenols extraction, as strengthened by the liquid chromatography and mass spectroscopy (LC-MS) and fourier transform infrared spectroscopy (FTIR) analysis. *M. esculenta* methanol extract showed the highest DPPH and ABTS radical scavenger antioxidant activity with IC_50_ values of 39.29 μg/mL and 52.83 μg/mL, respectively, while the ethyl acetate and aqueous extracts revealed minimum antioxidant potential. Methanol extract also revealed higher phenolic content, 88.94 ± 0.24 mg of equivalent gallic acid (GAE)/g), measured by the Folin–Ciocalteu method, while the minimum content was recorded for aqueous extract (62.38 ± 0.14 GAE/g). The highest flavonoid content was observed for methanol extract, 67.44 ± 0.14 mg quercetin equivalent (QE)/g) measured by an aluminum chloride colorimetric method, while the lowest content was recorded for aqueous extract (35.77 ± 0.14 QE/g). Antimicrobial activity findings also reveal that the methanol extract led to a higher inhibition zone against bacterial and fungal strains. FTIR analysis reveals the presence of various functional groups, viz. alkenes, amines, carboxylic acids, amides, esters, alcohols, phenols, ketones, carboxylic acids, and aromatic compounds. This FTIR analysis could serve as a basis for the authentication of *M. esculenta* extracts for future industrial applications. Compounds identified by LC-MS analysis were gallic acid, myricanol, myricanone, epigallocatechin 3-*O*-gallate, *β*-sitosterol, quercetin, *p*-coumaric acid, palmitic acid, *n*-pentadecanol, *n*-octadecanol, stigmasterol, oleanolic acid, *n*-hexadecanol, *cis*-*β*-caryophyllene, lupeol, and myresculoside. *Conclusion:* This study suggests that the methanolic extract from *M. esculenta* leaves has strong antioxidant potential and could be a significant source of natural antioxidants and antimicrobials for functional foods formulation.

## 1. Introduction

Natural products play an important role in both innovation and promotion of new drugs [1,2]. Almost 25% of conventional drugs contain phytocompounds extracted from higher plants [3]. As indicated by the World Health Organization (WHO), about 80% of the world’s population living in developing countries basically depends on plants for basic healthcare [4]. Botanicals of medicinal importance have been explored for their antioxidant potentialities [5]. 

Earlier studies have found that the antioxidant activity of several botanicals is mainly due to their richness in phenolic compounds, viz. flavonoids, phenolic acids, vitamins C and E, and various carotenoids [6,7]. These herbal antioxidants are very effective in preventing the destructive physiopathology triggered by oxidative stress due to free radicals’ overproduction [8]. 

Oxidative stress is caused by reactive oxygen species (ROS). ROS is a form of atmospheric oxygen which has been reduced to become singlet oxygen (O^−^). Generally, ROS formation involves the transfer of one or more electrons from O_2_ to form several types of radicals, including hydrogen peroxide (H_2_O_2_), hydroxyl radicals (HO^*^), and superoxide radicals (O_2_^−^). These oxygen radicals can do various types of damages leading to degenerative diseases, cancers, ulcers, and atherosclerosis [9]. Antioxidants play an important role in preventing oxidative damage; they neutralize free radicals, thus controlling chronic disease. However, synthetic antioxidants, such as butylatedhydroxytoluene (BHT) have been regulated as they have been suspected to cause liver damage and carcinogenesis [10]. This has prompted efforts to find alternative antioxidants from natural sources with fewer side effects. Pathology of various ailments, including carcinoma, cardiovascular and neurodegenerative disorders, hypertension, diabetes mellitus, and premature aging are associated with free radicals or ROS generation [11,12]. Environmental pollutants, radiation, chemicals, toxins, deep-fried foods, and spicy foods, as well as physical stress, are responsible for generating ROS, that induce the formation of abnormal proteins leading to antioxidants depletion in the immune system [13]. There are a number of endogenous antioxidant enzymes, viz. glutathione peroxidase, catalase, and superoxide dismutase, which are capable of deactivating free radicals and therefore maintaining optimal cellular functions [14]. However, endogenous antioxidants may not be sufficient to maintain optimal cellular functions under increased oxidative stress status, and therefore dietary antioxidants may be necessary [15]. 

Recently, researchers have focused on increasing human infections caused by bacteria and fungi. Medicinal plants also represent a rich source of antimicrobial agents. Since microorganisms have developed resistance to many antibiotics [16], the use of plant extracts and their isolated compounds has increased [17].

*Myrica esculenta* Buch.-Ham. ex. D. Don, named “Hairy Bayberry” and widely known as Kaiphal or Kataphala in the Indian subcontinent, is broadly used in Ayurveda (Indian traditional system of medicine) [18]. *Myrica* plants grow well in nitrogen depleted soils, mixed forests, agricultural, and marginal lands [19,20] *M. esculenta* is known for its edible fruits and other by-products. Indeed, its fruits have been a potential income-generating source for local tribes of Meghalaya and sub-Himalayan region [21,22]. The entire plant parts of *M. esculenta* have a huge nutritional and therapeutic importance. Indeed, the presence of distinct bioactive compounds, such as alkaloids, flavonoids, glycosides, tannins, terpenoids, saponins, and volatile oils has been increasingly reported as related to its pharmacological effects [23]. This species is fundamentally the same as *M. rubra*, which is ordinarily found in China and Japan. Though, *M. esculenta* fruits are smaller than about 4–5 mm compared to the *M. rubra* fruits (12–15 mm). Although the information on phenolic content and the antioxidant role of the fruit extract, juice, jam, and marc of *M. rubra* [24,25,26] is available, this information is lacking for *M. esculenta*. Earlier research reported the antimicrobial potential of *M. esculenta* fruit and bark [23,27,28,29], but still, no research has been carried out on the antimicrobial activity of its leaves. The possible differences in the composition and biological activities between extracts made by using different solvents on the extraction of plant natural products are well described [30].

Hence, there is a great need to explore the antioxidant properties of the species. Thus, the purpose of the present study was to explore the phenolic and flavonoid contents of *M. esculenta* leaves and to evaluate their antioxidant and antimicrobial activity using different *in vitro* models.

## 2. Materials and Methods

### 2.1. Chemicals and Reagents

2,2-diphenyl-1-picrylhydrazyl (DPPH), 2,2″-azino-bis(3-ethylbenzothiazoline-6-sulphonic) acid (ABTS), Folin–Ciocalteu reagent, sodium carbonate, aluminum chloride, potassium acetate, sodium acetic acid, glacial acetic acid, ascorbic acid, quercetin, and gallic acid were purchased from Sigma-Aldrich (St. Louis, Mo, USA). All chemicals were of analytical grade.

### 2.2. Plant Material 

*M. esculenta* leaves were collected from the outskirt area of Chail Chowk, Mandi, Himachal Pradesh. The plant was identified, authenticated, and certified (HIMCOSTE/HPSBB/7085) by Dr. Pankaj Sharma, Himachal Pradesh State Biodiversity Board, Shimla, India.

### 2.3. Preparation of Extracts

Firstly, the plant leaves were washed with water to remove dirt and other foreign matters were separated and shade dried. Dried leaves were then milled to a coarse powder and then passed over sieve No. 14. The obtained dried powdered leaves of *M. esculenta* (20 g) were placed in the tube of Soxhlet apparatus in the form of a thimble and extracted with various solvents, such as ethyl acetate, methanol, and water (300 mL) at 60–65 °C for 3–4 h. The obtained extracts, respectively, ethyl acetate (EAE), methanol (ME) and aqueous (AE) extracts, were filtered while hot and dried by evaporation using a rotary vacuum evaporator and the final dried extract samples were kept at low temperature in the fridge for further study. The residue obtained from each extract was dissolved in the same solvent for further analysis. 

### 2.4. Total Polyphenols and Flavonoid Contents

The total phenolic content (TPC) and flavonoid (TFC) content of each *M. esculenta* leaf extracts were determined using the earlier reported method [31]. TPC was expressed as mg of gallic acid equivalent (GAE) per 100 g of extract, while the TFC was expressed as mg of quercetin equivalents (QE) per 100 g.

### 2.5. Fourier Transform Infrared Spectroscopy (FTIR)

Functional groups and types of chemical bonds present in phytochemicals were identified by Fourier transform infrared spectroscopy (FTIR) analysis. Light absorbed wavelength was the prominent aspect of chemicals bonds, which can be seen through the interpreted spectrum. Compound chemical bonds can be deduced via absorption infrared spectrum. Each extract (8 mg) was loaded to Fourier transform infrared spectrophotometer (System 2000, Perkin Elmer, Wellesly, MD, USA) for functional group analysis. The IR peak absorbance (wave number, cm^−1^) was recorded in the range of 4000 cm^−1^ to 400 cm^−1^.

### 2.6. LC-MS Analysis

The polyphenols in different leaves extracts were evaluated by chromatographic technique, using the earlier reported method [32]. The chromatographic system consists of an Agilent 1100 series HPLC instrument (Santa Clara, USA) equipped with an MS detector. Analytical separation was carried out in a C18 column (4.6 mm × 100 mm × 5 μm, Agilent Technology) with a flow rate of 0.8/min with two mobile solvent phases (eluent A = 10 mM ammonium acetate and 1% acetic acid in water; eluent B = 1% acetic acid in methanol). The gradient elution was performed as follows: 0.3 min, 15–50% A; 3–5.5 min, 50–90% A; 5.5–9 min, 90% A; 9–9.5 min, 90–15% A, 9.5–10 min, 15% A. The sample injection volume was 20 μL and the temperature of the column was fixed at 40 °C. Compounds were identified with the mass spectra and Rt with the NIST library of standard compounds. 

### 2.7. Antioxidant Activity

#### 2.7.1. DPPH Radical-Scavenging Activity

The free radical scavenging capability of each extracts solution on the DPPH radical was determined as previously described [33]. ME, EAE, and AE solutions were prepared at different concentrations from 20 to 100 μg/mL. The DPPH radical solution (50 μM) was added to the solution of various plant extracts concentrations and standard ascorbic acid individually. The reaction mixtures were shaken thoroughly and kept in the dark for 30 min. The control solution was prepared by adding 2 mL of methanol with 2 mL of DPPH solution. The absorbance of all the reaction mixtures and control solution was measured at 517 nm. The percentage inhibition was calculated by the following formula:(1)$$\%\;Inhibition=\left[\left(\frac{AC517nm\;-AS517nm}{AC517nm}\right)\times 100\right]$$
where, AC is the absorbance of Control and AS is the absorbance of the Sample.

The graph was plotted between % inhibition and different concentrations of plant extracts and ascorbic acid and IC_50_ value was determined.

#### 2.7.2. ABTS Assay

The reducing power of the crude extracts was determined using the ABTS assay as described earlier [34]. ME, EAE, and AE solutions were prepared at varying concentrations from 20 to 100 μg/mL.

One milliliter of distilled dimethyl sulfoxide was mixed to 0.2 mL of varying concentrations of the samples and 0.16 mL ABTS solution was added to obtain a volume of 1.36 mL. The absorbance was analyzed spectrophotometrically, after 20 min at 734 nm with a UV spectrophotometer. Control remained without a sample. ABTS scavenging capacity of the ABTS was expressed as IC_50_ (μg/mL) and the percentage of inhibition was calculated using the following formula:ABTS scavenging activity (%) = (A_0_ − A_1_)/A_0_ × 100(2)
where, A_0_: absorbance of the control, A_1_: absorbance of the sample.

### 2.8. Antimicrobial Activity

#### 2.8.1. Microbial Strains

Four pathogenic bacterial strains were used for antibacterial screening of the *M. esculenta* leaves extracts. Two Gram-positive bacteria: *Bacillus subtilis* (ATCC7722), *Staphylococcus aureus* (ATCC 6538) and two Gram-negative bacteria: *Escherichia coli* (ATCC 25922) and *Pseudomonas fluorescens* (ATCC 13525) were tested. Two pathogenic fungal strains *Aspergillus niger* (ATCC 16404) and *Candida albicans* (MTCC 227) were used to access the antifungal activity of *M. esculenta* leaves extracts. These strains were from Microbial Type Culture Collection and Gene Bank, Institute of Microbial Technology, Chandigarh.

#### 2.8.2. Preparation of Culture Medium and Inoculation

For the antibacterial activity, 35 g nutrient agar and 10 g agar–agar were suspended in distilled water (1000 mL) and dissolved by boiling. Media and Petri dishes by autoclaving at 15 lbs pressure for 20 min. Under aseptic conditions, 20 mL of media was dispended into sterilized Petri dishes to yield a uniform depth of 6 mm after solidification of the medium; the bacterial cultures were inoculated by a spread plating technique. The concentration of the microbial suspension was adjusted to the 2.0 McFarland standard and 50 μL of each microorganism’s suspension was spread on an agar plate.

#### 2.8.3. Disc Application and Incubation

Discs of 6 mm diameter were prepared from Whatman No. 1 filter paper. They were sterilized by autoclaving and subsequently dried at 80 °C for an hour. The sterilized discs were immersed in respective *M. esculenta* extracts and dried for 3–5 min. After drying, discs were placed on nutrient agar surface with flamed forceps and gently press down to ensure contact with the agar surface. The discs were spaced apart enough to avoid both reflection waves from the edges of the petri dishes and overlapping rings of inhibition; finally, the petri dishes were incubated for 24 h at 37 °C in an inverted position. After 24 h, the diameter (mm) of the inhibition zone around each disc was measured. Antibacterial activities were indicated by a clear zone of growth inhibition (mm). A triplicate antibacterial assay was performed for each bacterial strain and for the different solvent extracts and standard drug [34].

#### 2.8.4. Evaluation of Antifungal Activity

ME, EAE and AE were tested for antifungal activity by agar disc diffusion method. Antifungal activity of all the respective extracts of *M. esculenta* was screened for the *in vitro* growth inhibitory activity against *A. niger* and *C. albicans*. The fungi were cultured in the czepadox broth medium. The sterilized medium taken in the sterilized petri dishes were inoculated with a spore suspension of *A. niger* and *C. albicans*. The filter paper discs were immersed in respective extracts. After drying, the discs were placed on the surface of the czepadox broth medium with flamed forceps and gently pressed down to ensure contact with the medium surface. The petri dishes were incubated at 28 °C, and after 48 h the inhibition zone that appeared around the disc in each plate was measured. To rule out the activity of solvent used in the preparation of extracts, solvents (methanol, ethyl acetate, and water) were used in the control plate.

### 2.9. Statistical Analysis

The results were expressed as mean ± standard error mean (SEM). Statistical analysis of the data was carried out using the Student’s *t*-test and the results were considered significant when *p* <0.05.

## 3. Results and Discussion 

### 3.1. Total Phenolic Content (TPC)

Total phenolic content is the method selected to determine the phenolic level in plant extracts. These phenolic compounds possess redox properties, which allow them to act as potential antioxidants [35,36]. As a basis, phenolic content was measured using the Folin–Ciocalteu reagent in each extract. The results were expressed as gallic acid equivalents (GAE) per gram of dry extract weight (Table 1). The results indicate that the ME exhibited higher TPC comparatively to the EAE and AE, which were about 88.94±0.24 mg GAE/g for ME, 75.83±0.19 mg GAE/g for EAE, and 62.38±0.14 mg GAE/g for AE. TPC was calculated using the following linear equation, based on the calibration curve of gallic acid (y = 0.006x + 0.459, R^2^ = 0.981).

The greater phenolic level in the ME may suggest higher bioactivity, viz. antioxidant and antimicrobial activities. Several studies have reported the relationship between phenolic content and antioxidant activity. Veliogluet al. [37,38] reported a strong relationship between the TPC and antioxidant activity in certain plant products. Sengul et al. [39] also reported that phenolic compounds serve in plant defense mechanisms to counteract ROS formation, promoting survival and preventing both molecular damages and microorganisms, insects, and herbivores attacks.

### 3.2. Total Flavonoid Content (TFC)

Flavonoids are secondary metabolites with antioxidant activity, the potency of which depends on the number and position of free OH groups [40]. As a basis, the quantitative determination of the flavonoid contents in selected plant extracts were determined using aluminum chloride in a colorimetric system. The outcomes were expressed as quercetin equivalents (QE) per gram of dry extract weight (Table 1). The results showed that the ME exhibited higher TFC as compared to the EAE and AE, being approximately about 67.44 ± 0.14 mg GAE/g for ME, 46.83 ± 0.19 mg QE/g for EAE, and 35.77 ± 0.14 mg GAE/g for AE. TFC was calculated using the following linear equation, based on the calibration curve of quercetin (y = 0.006x + 0.351, R^2^= 0.986).

Methanol is widely used to extract plant natural compounds, here, the richness in bioactive phytoconstituents found in *M. esculenta* leaves ME could explain the widespread therapeutic use of this folklore botanical [18].

Flavonoids are a major group of phenolic compounds widely distributed in plants. They play an important role in giving flavor and color to fruits and vegetables [41]. The presence of hydroxyl groups confers to flavonoids radical scavenging ability. The same concept as for TPC was applied in TFC determination, wherein the presence of a flavonoid-aluminum complex led to color change [42]. *M. esculenta* leaves ME exhibited the highest values for TPC and TFC, as shown in Table 1, indicating that, among the studied extracts, ME is the richest in phenolic compounds abundance. This presumption was supported by the antioxidant and cytotoxic activities found to ME. Thus, *M. esculenta* can be considered a promissory source of phenolic compounds compared with other natural sources of these compounds [23].

### 3.3. Fourier-Transform Infrared Spectroscopy (FTIR)

FTIR is an important method used in the authentication of plant materials and justifies the use of this method as an authentication tool in the present study, especially for comparison with future studies and/or future industrial applications using *M. esculenta* extracts [43]. The functional groups of compounds were examined by Fourier-transform infrared spectroscopic studies by their peak values (cm^−1^). Alkenes, amines, carboxylic acids, amides, esters, alcohols, phenols, ketones, carboxylic acids, and aromatic compounds were identified. Aromatics, sulfones, and aliphatic amines showed main peaks at 1460, 1365 and 1074 cm^−1^. Different intensity peaks were identified for primary and secondary amines at (3462, 1603,745, and 780 cm^−1^) carboxylic acids at (3300, 1712, and 1290 cm^−1^), alcohols and phenols at (3360, 3307, and 1290 cm^−1^), alkenes at (2965 and 1634 cm^−1^) (Table 2, Table 3 and Table 4).

### 3.4. LC-MS Analysis

The compounds identified by LC-MS analysis were presented in Table 5. A total of 18 compounds were identified in the different extracts of *M. esculenta* leaves, in particular phenolics, flavonoids, and arylheptanoids. Myricanol, myricanone, epigallocatechin 3-O-gallate, *β*-sitosterol, quercetin, *p*-coumaric acid, palmitic acid, *n*-pentadecanol, *n*-octadecanol, stigmasterol and oleanolic acid, *n*-hexadecanol, *cis-β*-caryophyllene, lupeol were identified in ME. Myresculoside was found in AL, while gallic acid was present in EAE. Earlier literature reveals the presence of epigallocatechin 3-*O*-gallate, myricetin, and stigmasterol in the bark, while *β*-sitosterol was found in leaves. Lupeol, oleanolic acid were identified in the bark and leaves of the plant. Gallic acid, *p*-coumaric acid, myricetin were reported in bark, leaves and fruits, while *n*-hexadecanol, myresculoside, *β*-caryophyllene, *n*-octadecanol, myricanol, and myricanone in bark, leaves and roots [44]. 

### 3.5. Antioxidant Activity

#### 3.5.1. DPPH Radical Scavenging Activity

DPPH radical scavenging activity is based on one-electron reduction, which represents the free radical reducing activity of antioxidants. Ascorbic acid (AA) was used as a positive control. The results (Figure 1) showed the DPPH radical inhibition percentage of *M. esculenta* crude extracts and standard at different concentrations. The lowest IC_50_ were detected for ME, followed by EAE and AE, with IC_50_ values of 39.29, 65.19, and 91.90 µg/mL, respectively (Table 6). The IC_50_ of ascorbic acid was 16.53 µg/mL. As the lower IC_50_ (concentration required for 50% inhibition) value possesses a higher antioxidant activity, ME was those with higher ability to scavenge free radicals compared to EAE and AE. The high antioxidant activity showed by ME had a positive relationship with TPC. Previous studies have shown that antioxidant capacity is highly associated with both TFC and TPC [45,46].

#### 3.5.2. 2,2′-azino-bis(3-ethylbenzothiazoline-6-sulphonic acid) assays

*M. esculenta* ME revealed a dose-dependent ABTS radical scavenging potential (Figure 2).

At 20 μg/mL, the percentage inhibition of the ME, EAE, and AE were 43.09%, 39.73%, and 36.19%, respectively, and that of ascorbic acid was 50.84%. The IC_50_ value of ascorbic acid was 16.51 μg/mL, whereas ME, EAE, and AE showed 52.83, 71.97, and 92.22 μg/mL, respectively (Table 6).

Antioxidants play an important role in preventing oxidative damage. They neutralize and break the free radical chains, thus controlling oxidative stress-initiated disease. However, unlike natural antioxidants, synthetic antioxidants, such as BHT, are controlled substances, as they are suspected to cause liver damage and carcinogenesis [47]. Thus, to minimize the negative side effects, natural sources should be investigated in an extensive manner to find effective and safer alternative antioxidants [48]. In the last decade, several plant-derived natural compounds have shown to be credible alternatives to these potentially harmful synthetic antioxidants, such as the lignan secoisolariciresinol [49]. 

Several phenolic compounds, viz. simple phenolics, phenolic acids, anthocyanins, and flavonoids present in plants have witnessed a great interest owing to their rich antioxidant potential, which includes free radicals scavenging, and ameliorating effects against mutagens, carcinomas, and inflammatory pathological processes [50,51]. Earlier studies reported that due to its redox properties, phenolics compounds also serve as reducing agents, hydrogen givers, singlet oxygen inhibitors and effective metal chelators [52].

In the ABTS assay, the ABTS^+^ radical scavenging capacity observed to ME was found to be very high. In the present study, there was greater DPPH radical scavenging activity by increasing the concentration of plant extracts, which could mean a greater ability to transfer a hydrogen atom, forming a lighter solution, proportional to the number of electrons obtained [53]. Thus, it may be suggested that owing to good hydrogen atom transferability, *M. esculenta* reduces the radical to the corresponding hydrazine and exhibits a DPPH scavenging role, although the ABTS^+^ scavenging role of extracts was found to be much smaller than that of DPPH radical activity. It has been observed that the stereoselectivity of the radicals and the solubility of extracts in varied solvent systems can affect the ability of extracts to interact and reduce various radicals. It has also been found that some compounds having ABTS^+^ scavenging activity did not show DPPH scavenging potential [54]. Indeed, ABTS and DPPH are two antioxidant assays that allow to access the radical scavenging activity of an extract. However, their strict mechanisms are not exactly identical. ABTS is based on a strict hydrogen atom transfer (HAT) mechanism, whereas DPPH, in addition to this HAT mechanism, could also evidence the presence of an antioxidant acting through an electron transfer (ET) mechanism [55]. So, here the slightly higher (but not significant) antioxidant activity determined with DPPH could suggest the presence of antioxidants acting through an ET mechanism, but the present results suggest that the main antioxidant contributors act through a HAT mechanism.

The relationship between phenolic compounds and antioxidant activity is well described, but here we provide new information on the possible mechanism using these two different radical scavenging assays, in relation to the relative phytochemical composition of each extract, thus far the most complete phytochemical analysis of this important folk medicinal plant. Moreover, it is well accepted that several factors, such as the phytochemical composition, closely related to the extraction method used, could greatly influence the antioxidant activity [56]. We used different extraction solvents to evaluate how it could influence the phytochemical composition as well as the antioxidant activity of *M. esculenta* extract. Here, the methanol extraction gives the best results both in terms of *M. esculenta* extract phytochemical diversity and antioxidant potential. For example, compared to the other extracts, ME was richer in phenolics, flavonoids, and arylheptanoid antioxidant compounds, such as *p*-coumaric acid, catechin, quercetin, and myricanone. The influence of extraction solvent in the antioxidant activity of the corresponding extracts have already been described for these compounds [55], and here a clear distinction between extracts composition was also observed, being ME richer in phenolics and arylheptanoids, known for their antimicrobial potential [18,57].

### 3.6. Antimicrobial Activity

The antibacterial and antifungal effects of *M. esculenta* leaf extracts are presented in Table 7, Table 8 and Table 9. Gentamicin was the standard drug used to assess the antibacterial potential, while fluconazole was used as a standard for antifungal activity determination. In the present study, ME was more active against *B. subtilis*, *S. aureus*, *E. coli* and *P. fluorescens*. *S. aureus* and *E. coli* are commonly involved in skin, delicate tissues, bones and joints infections, abscesses, as well as normal heart valves contamination in human beings [58]. *E. coli* causes serious urinary tract infections and is also a source of antibiotic-resistant genes transfer from infected food animals to humans [59]. The *M. esculenta* extract potential against *S. aureus* should also be explored to develop a topical antimicrobial therapy to promote skin wound healing. Parasitic diseases, especially those brought by *Candida* species are the most challenging contaminations faced by vulnerable patients, such as with HIV/AIDS patients. *M. esculenta* ME also exhibited good potential against *C. albicans*. 

Previous studies reported the antimicrobial effect of *M. esculenta* fruit extract. Mann et al. [32] evaluated the antimicrobial activity of *M. esculenta* fruit extract against three Gram-positive (*S. aureus*, *B. subtilis*, *Staphylococcus epidermis*) and three Gram-negative (*E. coli, Salmonella enterica,* and *Proteus mirabilis*) bacterial strains. The results of this study revealed that *S. epidermis* and *S. aureus* were more sensitive to methanolic extracts. The inhibition zones recorded for the Gram-positive strains *S. aureus*, *B. subtilis*, *S. epidermis* were, respectively, 16 ± 0.5, 12 ± 0.2, and 18 ± 0.5 mm, while for the Gram-negative strains *E. coli, S. enterica,* and *P. mirabilis* were, respectively, 15.5 ± 0.5, 10 ± 0.1, and 12 ± 0.2 mm. Tetracycline was used as positive control in the study. In the current study, the inhibition zones recorded for *M. esculenta* ME (at 100 μg/mL) against the Gram-positive strains *B. subtilis* and *S. aureus* were, respectively, 14.7 ± 0.11 and 13.14 ± 0.17 mm and for the Gram-negative strains *E. coli* and *P. fluorescens* were 12.01 ± 0.02 and 12.01 ± 0.05, respectively. 

Previously reported studies also show that plants exhibit its antimicrobial activity due to the presence of bioactive compounds, namely *p*-coumaric acid [60], saponins [61], oleanolic acid [62], and orstigmasterol [63]. Here, the presence of a high amount of arylheptanoid myricanone in ME, while absent in the other extracts, was of particular interest. This compound has been proposed to exhibit antioxidant and antimicrobial properties [18].

## 4. Conclusions

This study demonstrated that *M. esculenta* leaf extracts have interesting antioxidant and antimicrobial properties attributed to their richness in phytochemicals, such as phenolics and flavonoids. The TPC and TFC quantification, as well as the individual composition obtained by LC-MS for each *M. esculenta* extract (ME, EAE and AE) revealed the strong impact of extraction solvent selected on both bioactive compounds content and composition. *M. esculenta* ME was the richest in terms of TPC and TFC, and the LC-MS analysis revealed the presence of simple phenolics, flavonoids, and arylheptanoids. New information was also provided here on possible antioxidant mechanisms of *M. esculenta,* specifically acting as free radical scavenger. *M. esculenta* ME also revealed the strongest antimicrobial activity against both Gram-positive and Gram-negative bacteria and even fungi. The biological activity of the different *M. esculenta* extracts seems to be strongly related to the phenolic acids and flavonoids content. In particular, our results suggest that the arylheptanoid myricanone may be a serious candidate for developing antioxidant and/or antimicrobial agents from *M. esculenta* leaves. Thus, given the remarkable antioxidant effects of *M. esculenta* leaf extracts, its consumption should be further exploited, as this plant may play an important role in preventing several health disorders involving free radicals’ overproduction, viz. carcinoma, cardiovascular diseases, and premature aging. However, more in-depth research is needed on the isolation and individual characterization of bioactive compounds for the development of promissory foods and/or cosmetic preservatives.

## Figures and Tables

**Figure 1 biomolecules-09-00357-f001:**
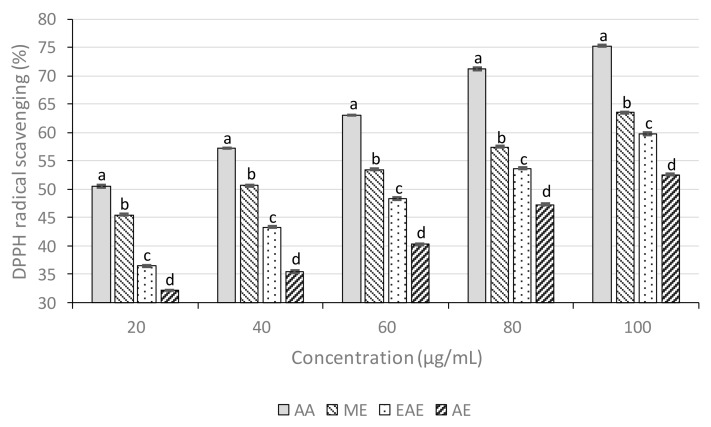
DPPH (2,2-diphenyl-1-picrylhydrazyl) radical scavenging activity of leaf extracts. Statistical significance was determined at *p* < 0.05 and is indicated with different letters (each concentration was evaluated independently).

**Figure 2 biomolecules-09-00357-f002:**
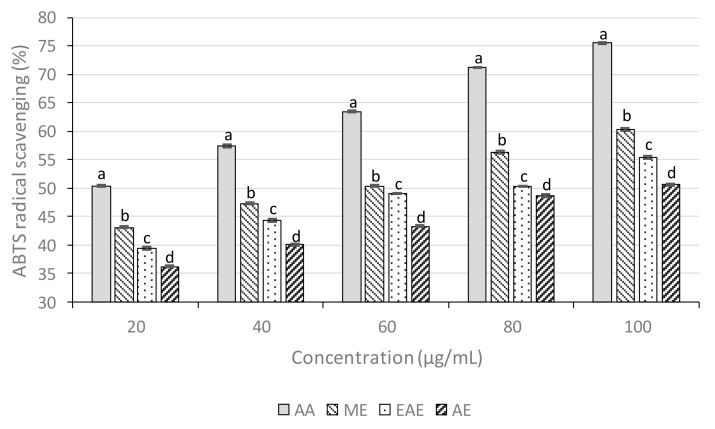
ABTS[2,2″-azino-bis(3-ethylbenzothiazoline-6-sulphonic) acid] radical scavenging activity of leaves extracts. Statistical significance was determined at *p* < 0.05 and is indicated with different letters (each concentration was evaluated independently).

**Table 1 biomolecules-09-00357-t001:** Total phenolic and flavonoid content.

Extracts	Phenolic Content (mg/g GAE)	Flavonoid (mg/g QE)
ME	88.94 ± 0.24 ^a^	67.44 ± 0.14 ^a^
EAE	75.83 ± 0.19 ^b^	46.83 ± 0.19 ^b^
AE	62.38 ± 0.14 ^c^	35.77 ± 0.14 ^c^

All values represent means ± SEM of three replicates. ME: Methanolic extract; EAE: Ethyl acetate extract; AE: Aqueous extract. Statistical significance was determined at *p* < 0.05 and is indicated with different letters.

**Table 2 biomolecules-09-00357-t002:** FTIR spectra of *M. esculenta* leaves methanolic extract.

S. No.	Peak (cm^−1^)	Frequency Range	Functional Groups
1	3462	3500–3100	N-H stretching vibration presence of primary and secondary amines and amides
2	3360	3500–3200	O-H stretching vibration presence of alcohols, phenols
3	3300	3300–2500	O-H stretching vibration presence of carboxylic acids
4	2965	3000–2850	C-H stretching vibration presence of alkenes
5	1712	1725–17051725–1700	C=O stretching vibration presence of ketone and carboxylic acid
6	1603	1640–1550	N-H bending vibration presence of primary and secondary amines and amides
7	1460	1500–1400	C-C stretching vibration presence of aromatics
8	1365	1375–1300	S=O stretching vibration presence of sulfones, sulfonyl chlorides, sulfates and sulfonamides
9	1290	1320–1000	C-O stretching vibration presence of alcohols, carboxylic acids, esters, ethers and anhydrides
10	1074	1250–1020	C-N stretching vibration presence of aliphatic amines
11	745,680	910–665	N-H stretching vibration presence of primary and secondary amines

**Table 3 biomolecules-09-00357-t003:** FTIR spectra of *M. esculenta* leaves ethyl acetate extract.

S. No.	Peak (cm^−1^)	Frequency Range	Functional Groups
1	3307	3500–3200	O-H stretching vibration presence of alcohols, phenols
2	1634	1680–1640	-C=C- stretching vibration presence of alkenes

**Table 4 biomolecules-09-00357-t004:** FTIR spectra of *M. esculenta* leaves aqueous extract.

S. No.	Peak (cm^−1^)	Frequency Range	Functional Groups
1	3307	3500–3200	O-H stretching vibration presence of alcohols, phenols
2	1634	1680–1640	-C=C- stretching vibration presence of alkenes

**Table 5 biomolecules-09-00357-t005:** LC-MS analysis of *M. esculenta* leaves extracts.

Compounds	Rt (min)	m/z	ME	EAE	AE
Epigallocatechin 3-*O*-gallate	8.55	458.37	++	−	−
Oleanolic acid	8.85	456.71	++	−	−
Eudesmol acetate	11.13	266.43	+++	−	−
n-pentadecanol	11.13	227.31, 227.98	+++	−	−
n-octadecanol	11.13	269.08, 269.34	+++	−	−
Stigmasterol	12.15	415.52	+++	−	−
Palmitic acid	12.15	257.21	++	−	−
*β*-sitosterol	12.15	415.52, 416.45	+++	−	−
n-hexadecanol	12.94	241.40	++	−	
Catechin	15.53	289.31	+++	−	−
*p*-coumaric acid	15.53	163.22, 163.78	+++	−	−
Myricanone	17.38	355.26	++	−	−
Quercetin	25.81	299.03	++	−	−
Myricanol	27.24	359.42	+	−	−
*β*-caryophyllene	41.68	205.10	−	−	+
Lupeol	41.70	424.40	−	−	+
Myresculoside	77.58	465.45	−	−	+
Gallic acid	95.78	171.25	−	+	−

ME: Methanolic extract; EAE: Ethyl acetate extract; AE: Aqueous extract.“+++” (relative high amount), “++” (relative medium amount), “+” (detected) and “−” (absent).

**Table 6 biomolecules-09-00357-t006:** IC_50_ values of *M. esculenta* extracts in DPPH and ABTS antioxidant assay.

Crude Extracts	DPPH Assay (µg/mL)	ABTS Assay (µg/mL)
Ascorbic acid	16.53 ± 0.25 ^a^	16.51 ± 0.24 ^a^
Methanol	39.29 ± 0.19 ^b^	52.83 ± 0.17 ^b^
Ethyl acetate	65.19 ± 0.18 ^c^	71.97 ± 0.15 ^c^
Aqueous	91.90 ± 0.21 ^d^	92.22 ± 0.17 ^d^

Statistical significance was determined at *p* < 0.05 and is indicated with different letters.

**Table 7 biomolecules-09-00357-t007:** Activity of different *M. esculenta* leaves extracts against Gram-positive bacteria.

Concentration of Extract (µg/mL)	Zone of Inhibition (mm)					
	*B. subtilis* (ATCC 7722)			*S. aureus* (ATCC 6538)		
	AE	EAE	ME	AE	EAE	ME
10	6.93 ± 0.02 ^d^	12.02 ± 0.03 ^c^	14.01 ± 0.01 ^b^	8.04 ± 0.01 ^d^	10.06 ± 0.02 ^c^	12.71 ± 0.03 ^b^
25	7.01 ± 0.00 ^d^	12.07 ± 0.04 ^c^	14.01 ± 0.23 ^b^	8.16 ± 0.03 ^d^	10.39 ± 0.03 ^c^	12.85 ± 0.02 ^b^
50	8.04 ± 0.02 ^d^	12.11 ± 0.02 ^c^	14.06 ± 0.12 ^b^	8.51 ± 0.02 ^d^	10.43 ± 0.02 ^c^	12.89 ± 0.01 ^b^
100	8.08 ± 0.02 ^d^	12.19 ± 0.02 ^c^	14.7 ± 0.11 ^b^	8.98 ± 0.02 ^d^	10.77 ± 0.05 ^c^	13.14 ± 0.17 ^b^
**Gentamicin (10)**	16.05 ± 0.01 ^a^			16.04 ± 0.02 ^a^		

AE: Aqueous extract; EAE: Ethyl acetate extract; ME: Methanolic extract. Statistical significance was determined at *p* < 0.05 and is indicated with different letters (each concentration was evaluated independently).

**Table 8 biomolecules-09-00357-t008:** Activity of different *M. esculenta* leaves extracts against Gram-negative bacteria.

Concentration of Extract (µg/mL)	Zone of Inhibition (mm)					
	*E. coli* (ATCC 25922)			*P. fluorescens* (ATCC 13525)		
	AE	EAE	ME	AE	EAE	ME
10	7.02 ± 0.00 ^d^	8.04 ± 0.01 ^c^	11.03 ± 0.01 ^b^	7.06 ± 0.01 ^d^	10.01 ± 0.00 ^c^	11.02 ± 0.01 ^b^
25	7.09 ± 0.02 ^d^	8.47 ± 0.01 ^c^	11.05 ± 0.02 ^b^	7.17 ± 0.02 ^d^	10.37 ± 0.01 ^c^	11.08 ± 0.12 ^b^
50	7.8 ± 0.01 ^d^	8.96 ± 0.03 ^c^	11.32 ± 0.29 ^b^	8.06 ± 0.02 ^d^	11.02 ± 0.01 ^c^	11.09 ± 0.15 ^b^
100	8.10 ± 0.02 ^d^	9.03 ± 0.00 ^c^	12.01 ± 0.02 ^b^	8.16 ± 0.02 ^d^	11.09 ± 0.01 ^c^	12.01 ± 0.05 ^b^
**Gentamicin (10)**	16.09 ± 0.15 ^a^			16.02 ± 0.13 ^a^		

AE: Aqueous extract; EAE: Ethyl acetate extract; ME: Methanolic extract. Statistical significance was determined at *p* < 0.05 and is indicated with different letters (each concentration was evaluated independently).

**Table 9 biomolecules-09-00357-t009:** Activity of different *M. esculenta* leaves extracts against fungal strains.

Concentration of Extract (µg/mL)	Zone of Inhibition (mm)					
	*A. niger* (ATCC 16404)			*C. albicans* (MTCC 227)		
	AE	EAE	ME	AE	EAE	ME
10	8.45 ± 0.02 ^d^	9.05 ± 0.01 ^c^	9.99 ± 0.01 ^b^	9.37 ± 0.01 ^d^	9.45 ± 0.02 ^c^	10.48 ± 0.01 ^b^
25	8.77 ± 0.01 ^d^	9.14 ± 0.02 ^c^	10.02 ± 0.02 ^b^	9.42 ± 0.01 ^d^	9.56 ± 0.01 ^c^	10.59 ± 0.02 ^b^
50	8.88 ± 0.02 ^d^	9.31 ± 0.01 ^c^	10.10 ± 0.01 ^b^	9.47 ± 0.02 ^d^	9.60 ± 0.01 ^c^	10.90 ± 0.01 ^b^
100	8.94 ± 0.01 ^d^	9.38 ± 0.01 ^c^	10.35 ± 0.02 ^b^	9.53 ± 0.02 ^d^	9.63 ± 0.01 ^c^	11.00 ± 0.01 ^b^
**Fluconazole (10)**	14.03 ± 0.01 ^a^			14.10 ± 0.01 ^a^		

AE: Aqueous extract; EAE: Ethyl acetate extract; ME: Methanolic extract. Statistical significance was determined at *p* < 0.05 and is indicated with different letters (each concentration was evaluated independently).

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
