# Peer review of "Phytochemical Composition, Antioxidant, and Antimicrobial Attributes of Different Solvent Extracts from Myrica esculenta Buch.-Ham. ex. D. Don Leaves"

_biomolecules, 2019, doi:10.3390/biom9080357_

Round 1
Reviewer 1 Report
1. In the introduction of this manuscript authors talk only about antioxidant activity, but not about antimicrobial effects. It is not clear why they decided to study it. In my opinion, there is too many fundamental information about radicals and not so much explanation about possible differences between extracts made by using different solvents.
2. Procedure of extract preparation is lacking a lot of necessary information, such as the ratio of raw material and solvent, extraction time, etc.
3. LC-MS analysis description, at least short procedure, should be provided.
4. Microbial culture seeding procedure should be described more thoroughly – how was microbial suspension made, what was the concentration, volume, was the bacterial growth temperature the same in all cases (bacteria and fungi), etc. What was the diameter of discs?
5. The description about statistical analysis should be provided. How many times have the experiments been performed?
6. I did not understand the importance of FTIR test. There is no discussion provided in this section, comparison with similar experiments from other researchers. As authors used LC-MS method, it should be much more useful to test the amounts of the main active substances in different extracts. Also, it is not shown which identified substances were found in which type of extract.
7. DPPH and APS assay results are given without standard deviations. How many times were those experiments performed? The finding that higher antioxidant activity was related to higher content of phenolic compounds, is not new.
8. No results from antimicrobial activity testing are provide, only a short paragraph. There are no mentioned tables 7, 8 and 9 in this manuscript.
9. In summary, it is not clear from results in this manuscript, what authors would like to tell that could be characteristic to their tested plant – Myrica esculent. Which substances in the composition of this plant could be of great importance, what is different/better compared to other well-known plants.
10. It is not clear why authors used different methods to establish antioxidant activity (ABTS and DPPH). I did not understand the explanation why different results were obtained by different methods, also there is no explanation why aqueous extract possessed a very similar activity in both (ABTS and DPPH) assays.
11. In lines 105-106 authors state that extracts were dried. There is no explanation in later described methodics how the extracts were prepared for further tests (were they dissolved in the same solvents or in different one, how these extracts were prepared for LC-MS analysis?).
Author Response
Original Reviewer 1 Report
1. In the introduction of this manuscript authors talk only about antioxidant activity, but not about antimicrobial effects. It is not clear why they decided to study it. In my opinion, there is too many fundamental information about radicals and not so much explanation about possible differences between extracts made by using different solvents.
2. Procedure of extract preparation is lacking a lot of necessary information, such as the ratio of raw material and solvent, extraction time, etc.
3. LC-MS analysis description, at least short procedure, should be provided.
4. Microbial culture seeding procedure should be described more thoroughly – how was microbial suspension made, what was the concentration, volume, was the bacterial growth temperature the same in all cases (bacteria and fungi), etc. What was the diameter of discs?
5. The description about statistical analysis should be provided. How many times have the experiments been performed?
6. I did not understand the importance of FTIR test. There is no discussion provided in this section, comparison with similar experiments from other researchers. As authors used LC-MS method, it should be much more useful to test the amounts of the main active substances in different extracts. Also, it is not shown which identified substances were found in which type of extract.
7. DPPH and APS assay results are given without standard deviations. How many times were those experiments performed? The finding that higher antioxidant activity was related to higher content of phenolic compounds, is not new.
8. No results from antimicrobial activity testing are provide, only a short paragraph. There are no mentioned tables 7, 8 and 9 in this manuscript.
9. In summary, it is not clear from results in this manuscript, what authors would like to tell that could be characteristic to their tested plant – Myrica esculent. Which substances in the composition of this plant could be of great importance, what is different/better compared to other well-known plants.
10. It is not clear why authors used different methods to establish antioxidant activity (ABTS and DPPH). I did not understand the explanation why different results were obtained by different methods, also there is no explanation why aqueous extract possessed a very similar activity in both (ABTS and DPPH) assays.
11. In lines 105-106 authors state that extracts were dried. There is no explanation in later described methodics how the extracts were prepared for further tests (were they dissolved in the same solvents or in different one, how these extracts were prepared for LC-MS analysis?).
Author Response to Reviewer 1
Point 1.In the introduction of this manuscript authors talk only about antioxidant activity, but not about antimicrobial effects. It is not clear why they decided to study it. In my opinion, there is too many fundamental information about radicals and not so much explanation about possible differences between extracts made by using different solvents.
Response 1:We are grateful for your time and constructive comments on our manuscript.
The antimicrobial effect of Myrica esculenta have been described and justified the evaluation of this activity for our extract. The reference Kabra et al. 2019, Journal of Ayurveda and Integrative Medicine 10(1), pp. 18-24 have been added in the introduction. The possible differences in the composition and biological activities between extracts made by using different solvents on the extraction of plant natural products is well described. An example is now cited in the introduction: Sultana et al Molecules 2009, 14, 2167-2180; doi:10.3390/molecules14062167.
Point 2.Procedure of extract preparation is lacking a lot of necessary information, such as the ratio of raw material and solvent, extraction time, etc.
Response 2. The required information has been added in the paragraph 2.3. Preparation of extracts of the Materials and Methods.
Point 3. LC-MS analysis description, at least short procedure, should be provided.
Response 3.The LC-MS procedure used is described in the reference 26 already cited in the MS. However, a short paragraph describing the LC-MS procedure have been added.
Point 4. Microbial culture seeding procedure should be described more thoroughly – how was microbial suspension made, what was the concentration, volume, was the bacterial growth temperature the same in all cases (bacteria and fungi), etc. What was the diameter of discs?
Response 4.Some of the information were already provided in the original version of the MS. However, we pay a particular attention to provide the additional information in the different paragraphs of the section 2.8. Antimicrobial activity of the Materials and Methods.
Point 5. The description about statistical analysis should be provided. How many times have the experiments been performed?
Response 5. The paragraph “2.9. Statistical analysis” mentioning the suggested information have been added in the Materials and Methods.
Point 6.I did not understand the importance of FTIR test. There is no discussion provided in this section, comparison with similar experiments from other researchers. As authors used LC-MS method, it should be much more useful to test the amounts of the main active substances in different extracts. Also, it is not shown which identified substances were found in which type of extract.
Response 6. First, FTIR is an important method use in the authentication of plant materials and justify the use of this method as an authentication tool in the present study, especially for comparison with future studies. We added this information and cited the reference Drouet et al. 2018 Cosmetics 2018, 5, 30; doi:10.3390/cosmetics5020030 for this purpose in the revised version of our MS.
Second, the information about the relative quantity for each identified compound in in each extract AA, ME, EAE and AE have been added in Table 5 using the notation “+++” (relative high amount), “++” (relative medium amount”, “+” (detected) and “-” (absent). Indeed, we prefer this annotation in absence of authentic standards for each compound that did not allow us to propose absolute quantitation for each individual compound.
Point 7. DPPH and APS assay results are given without standard deviations. How many times were those experiments performed? The finding that higher antioxidant activity was related to higher content of phenolic compounds, is not new.
Response 7. i) We have added the SD on the Figures 1 and 2 as well as the number of repetitions as suggested by Reviewer.
ii) We agree that the relation between phenolic compounds and antioxidant activity is not new, however here we also provided new information about the possible mechanism using 2 different radical scavenging activity (see answer to point 10), as well as the relative phytochemical composition of each extract (which constitute one of the most complete phytochemical analysis of this important traditional medicine plant). Moreover, it’s well accepted that several factors like the phytochemical composition closely related to the extraction method used could greatly influence the antioxidant activity. Here we used different extraction methods to evaluate how these extractions influenced the phytochemical composition as well as biological activities of M. esculenta extract. In addition, antimicrobial activity has been evaluated not only the antioxidant activity.
Point 8. No results from antimicrobial activity testing are provide, only a short paragraph. There are no mentioned tables 7, 8 and 9 in this manuscript.
Response 8.Sorry for this this forget. The Tables are now included in the revised version of our MS
Point 9. In summary, it is not clear from results in this manuscript, what authors would like to tell that could be characteristic to their tested plant – Myrica esculent. Which substances in the composition of this plant could be of great importance, what is different/better compared to other well-known plants.
Response 9.We have now answer point to point to the reviewer summary in the previous answer.
Point 10. It is not clear why authors used different methods to establish antioxidant activity (ABTS and DPPH). I did not understand the explanation why different results were obtained by different methods, also there is no explanation why aqueous extract possessed a very similar activity in both (ABTS and DPPH) assays.
Response 10.ABTS and DPPH are 2 assays that allow radical scavenging activity of an extract. However, their strict mechanisms are not exactly identical. Indeed, ABTS is based on a strict hydrogen atom transfer (HAT) mechanism, whereas DPPH, in addition to this HAT mechanism, could also evidence the presence of antioxidant acting through an electron transfer (ET) mechanism (Prior et al. 1998 J. Agric. Food Chem. 1998, 46, 2686−2693). So, here the slightly higher (but not significant) antioxidant activity determined with DPPH could suggest the presence of antioxidant acting trough an ET mechanism, but the present results suggest that the main antioxidant contributors act through a HAT mechanism. These points have been added and discussed in the revised version of our MS.
Point 11. In lines 105-106 authors state that extracts were dried. There is no explanation in later described methodics how the extracts were prepared for further tests (were they dissolved in the same solvents or in different one, how these extracts were prepared for LC-MS analysis?).
Response 11.As mentioned in point 2, the required information about extract preparation has been added in the paragraph 2.3. Preparation of extracts of the Materials and Methods.
Thanks for careful reading again.
Your inputs would surely improve the quality of our article.

Reviewer 2 Report
The article shows some results regarding the phenolics content and composition of three extracts from M. esculenta leaves, in addition, their antioxidant and antimicrobial properties were presented. The novel of this work is the phenolic composition in M. esculenta leaves, which may be potential bioactives. However, the authors should improve the results discussion before to the article publication. More details below:
Introduction:
Intro is suitable, however at the moment some studies about Myrica esculenta phenolic have been reported and should be cited in the article intro: Kabra et al. 2019, Journal of Ayurveda and Integrative Medicine 10(1), pp. 18-24; Kumar et al. 2017, Pharmacognosy Journal 9(6), pp. s103-s106; Bhatt et al. 2017, Food Chemistry 215, pp. 84-91; Goyal et al. 2013, Journal of Biosciences 38(4), pp. 797-803.
- Lines 54 and 56: radical’s formula are not correct; e.g. OH- is the formula of hydroxyl ion, whereas OH* is the formula of hydroxyl radical (see also O- and O2-)
- Line 62: to with?
Materials and methods:
- Section 2.3: I don’t understand the term “successively”, you obtained three crude extracts (ethyl acetate, MeOH and H2O) or three different fractions by means the increasing of solvent polarity, besides, Table 1 indicates “extracts”
- Section 2.6: please, provide a brief description of LC-MS analysis
Results and Discussion
- Line 185: “These herbal”
- Line 192: I consider that the word “ensures” is not suitable. Although phenolics are recognized as potent bioactives, Folin-Ciocalteu assay not ensures the presence of highly bioactive compounds
- Lines 207-208: the sentence is not clear; the methanol extract is widespread therapeutic use? Methanol is a very toxic solvent.
- Line 213: this sentence “flavonoid-aluminium complex causes a change in colour that directly indicates antioxidant activity” is not totally true
- Sections 3.1 and 3.2: Myrica esculenta can be considered as a promising source of phenolics compared with other natural sources of these compounds (e.g. Indian medicinal plants). The authors should have discussed it in the article
- Section 3.3: considering that the authors performed a suitable LC-MS analysis, the IR results are not necessary. I suggest remove from the article the IR results, they don’t provide a relevant information, in addition, the authors did not discuss these results.
- Table 5: how the compounds were identified? This information should be described in the section 2.6. Why some compounds have a m/z with three decimals, however, other have only one or two? You used different MS instruments with different resolutions? Again, the information of LC-MS from section 2.6
- Section 3.4: the phenolic composition results should be more discussed and related with both antioxidant and antimicrobial properties of Myrica esculenta extracts.
- Lines 252-253: you performed a correlation analysis? Please, cited the R2 value
- Section 3.5: what is the relationship between the phenolics identified in extracts and the antioxidants properties observed? Describe in the article
- What are the practical implications of the present work? Discussed in the article
- Section 3.6: Tables 7-9 were not presented in the article, hence, I cannot review this section
- Conclusion have be improve according to the R&D section changes
Author Response
Original Reviewer 2 Report
The article shows some results regarding the phenolics content and composition of three extracts from M. esculenta leaves, in addition, their antioxidant and antimicrobial properties were presented. The novel of this work is the phenolic composition in M. esculenta leaves, which may be potential bioactives. However, the authors should improve the results discussion before to the article publication. More details below:
Introduction:
Intro is suitable, however at the moment some studies about Myrica esculenta phenolic have been reported and should be cited in the article intro: Kabra et al. 2019, Journal of Ayurveda and Integrative Medicine 10(1), pp. 18-24; Kumar et al. 2017, Pharmacognosy Journal 9(6), pp. s103-s106; Bhatt et al. 2017, Food Chemistry 215, pp. 84-91; Goyal et al. 2013, Journal of Biosciences 38(4), pp. 797-803.
- Lines 54 and 56: radical’s formula are not correct; e.g. OH- is the formula of hydroxyl ion, whereas OH* is the formula of hydroxyl radical (see also O- and O2-)
- Line 62: to with?
Materials and methods:
- Section 2.3: I don’t understand the term “successively”, you obtained three crude extracts (ethyl acetate, MeOH and H2O) or three different fractions by means the increasing of solvent polarity, besides, Table 1 indicates “extracts”
- Section 2.6: please, provide a brief description of LC-MS analysis
Results and Discussion
- Line 185: “These herbal”
- Line 192: I consider that the word “ensures” is not suitable. Although phenolics are recognized as potent bioactives, Folin-Ciocalteu assay not ensures the presence of highly bioactive compounds
- Lines 207-208: the sentence is not clear; the methanol extract is widespread therapeutic use? Methanol is a very toxic solvent.
- Line 213: this sentence “flavonoid-aluminium complex causes a change in colour that directly indicates antioxidant activity” is not totally true
- Sections 3.1 and 3.2: Myrica esculenta can be considered as a promising source of phenolics compared with other natural sources of these compounds (e.g. Indian medicinal plants). The authors should have discussed it in the article
- Section 3.3: considering that the authors performed a suitable LC-MS analysis, the IR results are not necessary. I suggest remove from the article the IR results, they don’t provide a relevant information, in addition, the authors did not discuss these results.
- Table 5: how the compounds were identified? This information should be described in the section 2.6. Why some compounds have a m/z with three decimals, however, other have only one or two? You used different MS instruments with different resolutions? Again, the information of LC-MS from section 2.6
- Section 3.4: the phenolic composition results should be more discussed and related with both antioxidant and antimicrobial properties of Myrica esculenta extracts.
- Lines 252-253: you performed a correlation analysis? Please, cited the R2 value
- Section 3.5: what is the relationship between the phenolics identified in extracts and the antioxidants properties observed? Describe in the article
- What are the practical implications of the present work? Discussed in the article
- Section 3.6: Tables 7-9 were not presented in the article, hence, I cannot review this section
- Conclusion have be improve according to the R&D section changes
Author Response to Reviewer 2
Point 1. The article shows some results regarding the phenolics content and composition of three extracts from M. esculenta leaves, in addition, their antioxidant and antimicrobial properties were presented. The novel of this work is the phenolic composition in M. esculenta leaves, which may be potential bioactives. However, the authors should improve the results discussion before to the article publication. More details below:
Response 1.We now provide a more in-depth point to point discussion of our results as well as highlight the novelty. These additions have been highlighted in green in the revised version of our MS.
Point 2.Introduction:
Intro is suitable, however at the moment some studies about Myrica esculenta phenolic have been reported and should be cited in the article intro: Kabra et al. 2019, Journal of Ayurveda and Integrative Medicine 10(1), pp. 18-24; Kumar et al. 2017, Pharmacognosy Journal 9(6), pp. s103-s106; Bhatt et al. 2017, Food Chemistry 215, pp. 84-91; Goyal et al. 2013, Journal of Biosciences 38(4), pp. 797-803.
Response 2. These references have been added in the revised version of our MS.
Point 3.- Lines 54 and 56: radical’s formula are not correct; e.g. OH- is the formula of hydroxyl ion, whereas OH* is the formula of hydroxyl radical (see also O- and O2-)
- Line 62: to with?
Response 3.Lines 54 and 56: The radical formula has been corrected accordingly. Line 62: “to” has been deleted.
Point 4.Materials and methods:
- Section 2.3: I don’t understand the term “successively”, you obtained three crude extracts (ethyl acetate, MeOH and H2O) or three different fractions by means the increasing of solvent polarity, besides, Table 1 indicates “extracts”
Response 4.The section 2.3 have been rewritten and “extracts have been added in Table 1.
Point 5.Section 2.6: please, provide a brief description of LC-MS analysis.
Response 5.The LC-MS procedure used is described in the reference 26 already cited in the MS. However, a short paragraph describing the LC-MS procedure have been added.
Point 6.Results and Discussion
- Line 185: “These herbal”
Response 6. It has been corrected in the revised version of our MS.
Point 7.Line 192: I consider that the word “ensures” is not suitable. Although phenolics are recognized as potent bioactives, Folin-Ciocalteu assay not ensures the presence of highly bioactive compounds.
Response 7.It has been corrected in the revised version of our MS “could suggestensures a potent bioactivity and potential antioxidant and antimicrobial activities” was used instead.
Point 8. Lines 207-208: the sentence is not clear; the methanol extract is widespread therapeutic use? Methanol is a very toxic solvent.
Response 8.No, methanol is widely used to extract plant natural products. Note that following extraction methanol is evaporated so no present anymore for possible therapeutic uses.
Point 9. - Line 213: this sentence “flavonoid-aluminium complex causes a change in colour that directly indicates antioxidant activity” is not totally true.
Response 9.We delete this assertion that was mentioned by the Authors of the mentioned reference.
Point 10. Sections 3.1 and 3.2: Myrica esculenta can be considered as a promising source of phenolics compared with other natural sources of these compounds (e.g. Indian medicinal plants). The authors should have discussed it in the article.
Response 10.A reference has been added for this assertion.
Point 11.Section 3.3: considering that the authors performed a suitable LC-MS analysis, the IR results are not necessary. I suggest remove from the article the IR results, they don’t provide a relevant information, in addition, the authors did not discuss these results.
Response 11.First, FTIR is an important method use in the authentication of plant materials and justify the use of this method as an authentication tool in the present study, especially for comparison with future studies. We added this information and cited the reference Drouet et al. 2018 Cosmetics 2018, 5, 30; doi:10.3390/cosmetics5020030 for this purpose in the revised version of our MS.
Point 12.Table 5: how the compounds were identified? This information should be described in the section 2.6. Why some compounds have a m/z with three decimals, however, other have only one or two? You used different MS instruments with different resolutions? Again, the information of LC-MS from section 2.6
Response 12.The LC-MS procedure used is described in the reference 26 already cited in the MS. However, a short paragraph describing the LC-MS procedure have been added. The number of decimals has been homogenized.
Point 13.Section 3.4: the phenolic composition results should be more discussed and related with both antioxidant and antimicrobial properties of Myrica esculenta extracts.
- Lines 252-253: you performed a correlation analysis? Please, cited the R2 value.
Response 13.This information is now provided in the revised version of our MS.
Point 14.- Section 3.5: what is the relationship between the phenolics identified in extracts and the antioxidants properties observed? Describe in the article
Response 14.The information about the relative quantity for each identified compound in in each extract AA, ME, EAE and AE have been added in Table 5 using the notation “+++” (relative high amount), “++” (relative medium amount”, “+” (detected) and “-” (absent). Here we prefer this annotation in absence of authentic standards for each compound that did not allow us to propose absolute quantitation for each individual compound.
This now allow an easier evaluation of the relation between phenolics and antioxidant and antimicrobial activities. A sentence has been added to link each biological activity to some of these compounds.
Point 15.What are the practical implications of the present work? Discussed in the article
Response 15.Here we provided new information about the possible antioxidant mechanism of M. esculenta using 2 different radical scavenging activity (see answer to point 10), as well as the relative phytochemical composition of each extract (which constitute one of the most complete phytochemical analysis of this important traditional medicine plant). Indeed, ABTS and DPPH are 2 assays that allow radical scavenging activity of an extract. However, their strict mechanisms are not exactly identical. Indeed, ABTS is based on a strict hydrogen atom transfer (HAT) mechanism, whereas DPPH, in addition to this HAT mechanism, could also evidence the presence of antioxidant acting through an electron transfer (ET) mechanism (Prior et al. 1998 J. Agric. Food Chem. 1998, 46, 2686−2693). So, here the slightly higher (but not significant) antioxidant activity determined with DPPH could suggest the presence of antioxidant acting through an ET mechanism, but the present results suggest that the main antioxidant contributors act through a HAT mechanism. These points have been added and discussed in the revised version of our MS. Moreover, it’s well accepted that several factors like the phytochemical composition closely related to the extraction method used could greatly influence the antioxidant activity. Here we used different extraction methods to evaluate how these extractions influenced the phytochemical composition as well as biological activities of M. esculenta extract. In addition, antimicrobial activity has been evaluated not only the antioxidant activity.
Point 16.- Section 3.6: Tables 7-9 were not presented in the article, hence, I cannot review this section.
Response 16.Sorry for this this forget. The Tables are now included in the revised version of our MS.
Point 17.- Conclusion have be improve according to the R&D section changes
Response 17. : The conclusion have been rewritten accordingly in order to highlight the main results of the present work as mentioned earlier to Reviewer 2 about the novelty of this work:
Here we provided new information about the possible antioxidant mechanism of M. esculenta using 2 different radical scavenging activity (see answer to point 10), as well as the relative phytochemical composition of each extract (which constitute one of the most complete phytochemical analysis of this important traditional medicine plant). Indeed, ABTS and DPPH are 2 assays that allow radical scavenging activity of an extract. However, their strict mechanisms are not exactly identical. Indeed, ABTS is based on a strict hydrogen atom transfer (HAT) mechanism, whereas DPPH, in addition to this HAT mechanism, could also evidence the presence of antioxidant acting through an electron transfer (ET) mechanism (Prior et al. 1998 J. Agric. Food Chem. 1998, 46, 2686−2693). So, here the slightly higher (but not significant) antioxidant activity determined with DPPH could suggest the presence of antioxidant acting through an ET mechanism, but the present results suggest that the main antioxidant contributors act through a HAT mechanism. These points have been added and discussed in the revised version of our MS. Moreover, it’s well accepted that several factors like the phytochemical composition closely related to the extraction method used could greatly influence the antioxidant activity. Here we used different extraction methods to evaluate how these extractions influenced the phytochemical composition as well as biological activities of M. esculenta extract. In addition, antimicrobial activity has been evaluated not only the antioxidant activity.
Thanks for your careful reading and valuable suggestions. Your inputs would surely improve the quality of our article.

Round 2
Reviewer 1 Report
Authors added needed information regarding antimicrobial studies of Myrica esculenta in the Introduction, but the study aim description addresses only antioxidant activity studies and this does not correlate with the title of the manuscript. The antimicrobial activity studies seem to be as something additional but not systematically nicely incorporated into this manuscript. Antimicrobial activity is not mentioned also in conclusions of the study.
Tables with the antimicrobial activity data have been provided (table 7, 8 and 9), but there is no discussion related to the composition of extracts, different types of microorganisms, and there is no proper comparison with the data from other similar studies.
Authors provided a section “Statistical analysis”, but there is no information provided which statistical method (not a software program has been used for evaluation a statistically significant difference.
SD values in figures 1 and 2 seem quite strange, as they are shown both horizontally and vertically. Also, it seems it could be technical repetition of experiments, as all the vaues seem almost the same, and possibly could come not from the different extract batches.
Authors should clarify how the extracts were prepared for further tests in the section 2.3. Were they dissolved in the same solvents or in different one before further analysis?
There is no information provided at which temperature antifungal activity experiments have been done.
Author Response
Original Reviewer 1 Report
1. Authors added needed information regarding antimicrobial studies of Myrica esculenta in the Introduction, but the study aim description addresses only antioxidant activity studies and this does not correlate with the title of the manuscript. The antimicrobial activity studies seem to be as something additional but not systematically nicely incorporated into this manuscript. Antimicrobial activity is not mentioned also in conclusions of the study.
2. Tables with the antimicrobial activity data have been provided (table 7, 8 and 9), but there is no discussion related to the composition of extracts, different types of microorganisms, and there is no proper comparison with the data from other similar studies.
3. Authors provided a section “Statistical analysis”, but there is no information provided which statistical method (not a software program has been used for evaluation a statistically significant difference.
4. SD values in figures 1 and 2 seem quite strange, as they are shown both horizontally and vertically. Also, it seems it could be technical repetition of experiments, as all the values seem almost the same, and possibly could come not from the different extract batches.
5. Authors should clarify how the extracts were prepared for further tests in the section 2.3. Were they dissolved in the same solvents or in different one before further analysis?
6. There is no information provided at which temperature antifungal activity experiments have been done.
Author Response
Point 1. Authors added needed information regarding antimicrobial studies of Myrica esculenta in the Introduction, but the study aim description addresses only antioxidant activity studies and this does not correlate with the title of the manuscript. The antimicrobial activity studies seem to be as something additional but not systematically nicely incorporated into this manuscript. Antimicrobial activity is not mentioned also in conclusions of the study.
Response 1: We are grateful for your time and constructive comments on our manuscript.
We have improved the aim and conclusion of our MS as per your suggestion.
2. Tables with the antimicrobial activity data have been provided (table 7, 8 and 9), but there is no discussion related to the composition of extracts, different types of microorganisms, and there is no proper comparison with the data from other similar studies.
Response 2: Thank you for pointing this out. We have incorporated the earlier reported antimicrobial study results of M. esculenta fruits in the discussion (section 3.6) line no. 353 – 363 of MS.
Point 3. Authors provided a section “Statistical analysis”, but there is no information provided which statistical method (not a software program has been used for evaluation a statistically significant difference.
Response 3: We have revised the section 2.9 “Statistical analysis” in RMS.
Point 4. SD values in figures 1 and 2 seem quite strange, as they are shown both horizontally and vertically. Also, it seems it could be technical repetition of experiments, as all the values seem almost the same, and possibly could come not from the different extract batches.
Response 4: The values present in figures 1 and 2 are SEM value but mistakenly we have written SD. All the reading shown in figures come from different extract batches.
Point 5. Authors should clarify how the extracts were prepared for further tests in the section 2.3. Were they dissolved in the same solvents or in different one before further analysis?
Response 5: Thanks for your valuable suggestion. For further study extracts were dissolved in the same solvent, mentioned in the line no. 114 and 115 of revised MS.
Point 6. There is no information provided at which temperature antifungal activity experiments have been done.
Response 6: We have mentioned about the temperature in line no. 197 of revised MS.
Reviewer 2 Report
According the author responses, bellow my comments:
Introduction:
Intro was complemented suitably
Materials and methods:
- Section 2.3: OK
- Section 2.6: it is not suitable, should be improve
* line 131: column characteristics? e.g. length, particle size, supplier
* Elution conditions?
* MS analysis conditions?
* How the compounds were identified?
Results and Discussion
- Line 185: OK
- Line 192: OK
- Lines 207-208 (228 RMS): the answer is not satisfactory; methanol is not a GRAS solvent. Author should change this sentence
- Line 213: OK
- Sections 3.1 and 3.2: the answer is not satisfactory, discussion did not improve
- Section 3.3: OK
- Table 5: the MS analysis method is not described. In addition, the decimals are not homogenous (see oleanolic acid)
- Section 3.4 and 3.5: what is the relationship between the phenolic compounds identified in the extracts (e.g. some specific compounds) and both antioxidant and antimicrobial activities observed?
Author Response
Original Reviewer 2 Report
Introduction:
Intro was complemented suitably
Materials and methods:
- Section 2.3: OK
- Section 2.6: it is not suitable, should be improve
* line 131: column characteristics? e.g. length, particle size, supplier
* Elution conditions?
* MS analysis conditions?
* How the compounds were identified?
Results and Discussion
- Line 185: OK
- Line 192: OK
- Lines 207-208 (228 RMS): the answer is not satisfactory; methanol is not a GRAS solvent. Author should change this sentence
- Line 213: OK
- Sections 3.1 and 3.2: the answer is not satisfactory, discussion did not improve
- Section 3.3: OK
- Table 5: the MS analysis method is not described. In addition, the decimals are not homogenous (see oleanolic acid)
- Section 3.4 and 3.5: what is the relationship between the phenolic compounds identified in the extracts (e.g. some specific compounds) and both antioxidant and antimicrobial activities observed?
Author Response:
Introduction:
Point 1: Intro was complemented suitably
Response 1: Thank you for your careful reading and valuable comments.
We are grateful for your time and positive comments on our manuscript.
Materials and methods:
Point 2:- Section 2.3: OK
Response 2: Thank you for positive comments on our manuscript.
Point 3:- Section 2.6: it is not suitable, should be improve
* line 131: column characteristics? e.g. length, particle size, supplier
* Elution conditions?
* MS analysis conditions?
* How the compounds were identified?
Response 3: Thanks for your suggestions and constructive comments.
We have incorporated the points suggested by you in section 2.6 of our MS.
Results and Discussion
Point 4: - Line 185: OK
Response 4: Thank you for positive comments on our MS.
Point 5: - Line 192: OK
Response 5: Thank you for positive comments on our MS.
Point 6:- Lines 207-208 (228 RMS): the answer is not satisfactory; methanol is not a GRAS solvent. Author should change this sentence.
Response 6: We agree that methanol is not a GRAS solvent, but have a greater extraction capacity than other solvents toward the extraction of plant secondary metabolites. That’s the reason why methanol is classically used as a solvent for metabolomics. Here our objectives are to study the antioxidant and antimicrobial activities of the extracts not to propose a new extract for the market. Moreover, following extraction methanol can be evaporated and the extract resuspended in a GRAS solvent.
Earlier research also reveals that the greater effectiveness of the methanol extracts resulted from the fact that methanol is a very useful solvent capable of extracting a wide range of compounds. [Ali-Shtayeh et al.1999].
Point 7: - Line 213: OK
Response 7: Thank you for positive comments on our MS.
Point 8: - Sections 3.1 and 3.2: the answer is not satisfactory, discussion did not improve.
Response 8: We have improved these sections.
Point 9:- Section 3.3: OK
Response 9: Thank you for positive comments on our MS.
Point 10:- Table 5: the MS analysis method is not described. In addition, the decimals are not homogenous (see oleanolic acid)
Response 10: We have improved the MS analysis methodology section 2.6 in the revised version of our MS.
Point 11: - Section 3.4 and 3.5: what is the relationship between the phenolic compounds identified in the extracts (e.g. some specific compounds) and both antioxidant and antimicrobial activities observed?
Response 11: Thanks for pointing this out.
As per your suggestion two paragraphs were added to discuss these points:
Line 339-343 for antioxidant: “Here, methanolic extraction give the best results both in terms of phytochemical diversity and antioxidant potential of the M. esulenta extract. For example, as compared to the other extracts, the ME is rich in the antioxidant compounds such as catechin, p-coumaric acid and quercetin. The influence of extraction solvent and antioxidant activity of the corresponding extracts have already been described for these compounds (Zhao et al. 2006).”
Line 339-343 for antimicrobial: “Once again a clear distinction between extracts composition was observed, ME being richer in flavonoids and phenolics known for their antimicrobial actions (Gutiérrez-Grijalva et al., 2018).”
All the Thanks for careful reading again.
Your inputs would surely improve the quality of our article.
Round 3
Reviewer 1 Report
Provided information about antimicrobial activity did not convinced me that antimicrobial studies were naturally incorporated in this manuscript, as there is no discussion how antimicrobial activity is related to antioxidant activity and also with the constituents found in the extracts.
Authors revised a section “Statistical analysis” and explained that they used t-test. However, in the whole manuscript I did not find the data statistically evaluated by comparing if there statistically significant differences between them. Authors tested the composition of extracts, but also there is no data about correlation between substances and tested activities.
Authors provided SEM for some data in figures 1 and 2, but not for all of them. Those provided SEM seem quite small as for different batches of extracts. Also, it is not shown if there is a statistically significant difference between obtained data.
Overall, as mentioned in my previous reviews, mostly I lack the rational background of the research described in this manuscript. All the discussion seems quite superficial for me. Authors tested the composition of extracts, but without analytical standards, thus the results are not accurate. Moreover, when discussing about tested biological activities, they do not try to find the relationships between constituents and activity results, mostly rely on the total amount of phenolic compounds and flavonoids and literature data, what is already known. I am lacking more detailed focus on different fractions of extracts, maybe specific compounds, at least more accurate quantitative analysis.
Author Response
Original Reviewer 1 Report
Provided information about antimicrobial activity did not convinced me that antimicrobial studies were naturally incorporated in this manuscript, as there is no discussion how antimicrobial activity is related to antioxidant activity and also with the constituents found in the extracts. Authors revised a section “Statistical analysis” and explained that they used t-test. However, in the whole manuscript I did not find the data statistically evaluated by comparing if there statistically significant differences between them. Authors tested the composition of extracts, but also there is no data about correlation between substances and tested activities. Authors provided SEM for some data in figures 1 and 2, but not for all of them. Those provided SEM seem quite small as for different batches of extracts. Also, it is not shown if there is a statistically significant difference between obtained data. Overall, as mentioned in my previous reviews, mostly I lack the rational background of the research described in this manuscript. All the discussion seems quite superficial for me. Authors tested the composition of extracts, but without analytical standards, thus the results are not accurate. Moreover, when discussing about tested biological activities, they do not try to find the relationships between constituents and activity results, mostly rely on the total amount of phenolic compounds and flavonoids and literature data, what is already known. I am lacking more detailed focus on different fractions of extracts, maybe specific compounds, at least more accurate.
Authors Responses
Point 1.Provided information about antimicrobial activity did not convinced me that antimicrobial studies were naturally incorporated in this manuscript, as there is no discussion how antimicrobial activity is related to antioxidant activity and also with the constituents found in the extracts.
Response 1: We are grateful for your time and constructive comments on our manuscript.
We have improved the discussion section of our revised MS (line no. 362-375). In particular these additions are highlighted in blue.
Point 2.Authors revised a section “Statistical analysis” and explained that they used t-test. However, in the whole manuscript I did not find the data statistically evaluated by comparing if there statistically significant differences between them. Authors tested the composition of extracts, but also there is no data about correlation between substances and tested activities.
Response 2.
Thank you very much for your remark. We have included statistical analysis results on the Figures 1 and 2, as well as Tables 1, 6, 7 and 8. Statistical significance was determined at P < 0.05 and is indicated with different letters.
Point 3.Authors provided SEM for some data in figures 1 and 2, but not for all of them. Those provided SEM seem quite small as for different batches of extracts. Also, it is not shown if there is a statistically significant difference between obtained data.
Response 3.
We redrew these 2 figures as histograms. We have included statistical analysis results on these two figures 1 and 2. Statistical significance was determined at P < 0.05 and is indicated with different letters.
Point 4.Overall, as mentioned in my previous reviews, mostly I lack the rational background of the research described in this manuscript. All the discussion seems quite superficial for me. Authors tested the composition of extracts, but without analytical standards, thus the results are not accurate. Moreover, when discussing about tested biological activities, they do not try to find the relationships between constituents and activity results, mostly rely on the total amount of phenolic compounds and flavonoids and literature data, what is already known. I am lacking more detailed focus on different fractions of extracts, maybe specific compounds, at least more accurate.
Response 4. Thanks for pointing this out. As we have improved the manuscript discussion section and also revised the conclusion sectionof our manuscript to take this point into account. In particular these additions are highlighted in blue.

Reviewer 2 Report
The MS was improved by a suitable way. I consider that it can be accepted
Author Response
Original Reviewer 2 Report
The MS was improved by a suitable way. I consider that it can be accepted
Point 1. The MS was improved by a suitable way. I consider that it can be accepted
Response 1. Thank you very much for positive comment.
Authors are grateful for your time and suggestions on our manuscript.
